# EAGLE: Early Approximated Gradient-based Learning-rate Estimator

## Abstract

Improving the learning efficiency of deep learning models remains a significant research focus. In this paper, we propose **EAGLE (Early Approximated Gradient-based Learning-rate Estimator)**, a novel optimization method that accelerates parameter optimization. Firstly, to achieve faster loss convergence, EAGLE possesses the unique parameter update rule that leverages the local curvature of the loss landscape, derived from gradient variations between consecutive training steps. Secondly, to enhance training stability, it introduces a branching mechanism that adaptively switches to the existing Adam update rule under specific conditions where the EAGLE update rule might become unstable (e.g., extremely small gradient differences or locally upward convex shapes). In experiments on the GLUE SST-2 text classification task using a pre-trained GPT-2 model, EAGLE reached respectively the SGD with momentum's final loss value **6.83× faster** and the Adam's final loss value **6.77× faster**. Similarly, on the CIFAR-10 image classification task using a pre-trained ViT-B/16 model, EAGLE reached respectively the SGD with momentum's final loss value **3.41× faster** and the Adam's final loss value **6.60× faster**. To ensure reproducibility and promote further improvements, our code is publicly available on GitHub: https://github.com/keiotakmin/EAGLE

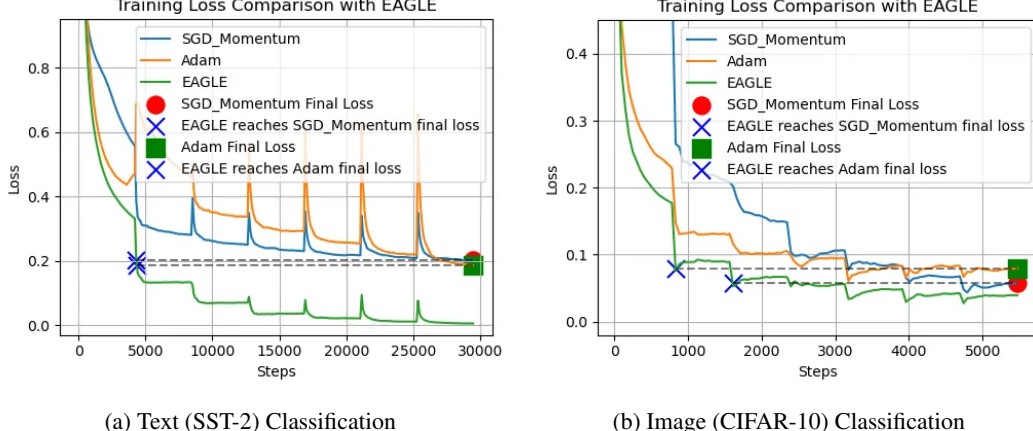

(a) Text (SST-2) Classification      (b) Image (CIFAR-10) Classification

Figure 1: Training Loss vs. Steps: EAGLE vs. Baseline Optimizers.

# 1    Intrduction

**Background**   In the field of optimization, second-order methods such as Newton's method and quasi-Newton methods have been extensively studied and their application to deep learning has been attempted. These methods approximate the objective function as a quadratic function by utilizing the Hessian matrix or its approximations, enabling parameter updates that incorporate curvature information. For example, on a simple quadratic function, they can reach the optimum in a single step[1]. Even in more complex landscapes, they can approach minima via more direct paths[2]. Consequently, they are theoretically expected to converge faster than first-order methods [3][4].

**Challenges**   Despite these theoretical advantages, Newton's method remains impractical for training large-scale models in modern deep learning. Computing and storing the full Hessian matrix incurs $\mathcal{O}(N^2)$ memory and computation costs for $N$ parameters[5], which is prohibitive when modern networks contain millions to billions of parameters. While quasi-Newton methods like L-BFGS[6] reduce these costs by maintaining only a small window of past gradient–parameter pairs, their applicability in deep learning is limited: most empirical studies focus on fully connected layers, and extensions to other architectures like CNNs[7] and RNNs[8] remain underexplored. For these reasons, the promise of "curvature-aware" fast convergence has not been fully realized in practical deep learning.

**Practicality of First-Order Methods**   As a result, first-order methods with high scalability and practicality have become the mainstream in modern deep learning. Stochastic Gradient Descent: SGD[9] remains a fundamental approach, achieving success in large-scale learning due to its computational and memory efficiency, and compatibility with mini-batch learning and momentum[10][11]. Adaptive methods such as Adam[12] have been emphasized for their suitability in large-scale problems involving massive datasets or model parameters. Extensions of Adam, including learning rate warmup and weight decay, have been shown to accelerate convergence in Transformer training[13] and enable stable large-scale pretraining of models like BERT[14]. However, these first-order optimization methods do not explicitly utilize curvature information, typically requiring many iterations to reach satisfactory convergence.

**Contributions**   Based on these background and challenges, we propose a new optimization method: EAGLE (early approximated gradient-based learning-rate estimator). EAGLE aims to merge the practicality of first-order methods with the fast convergence potential of second-order methods, resulting in accelerating parameter optimization.

Our main contributions of this research are as follows:

- Clarifying the gap between the theoretical appeal of fast convergence from second-order methods and the practical limitations

- Proposing a novel optimization algorithm designed to benefit from curvature information in a practical form

- Demonstrating that EAGLE exhibits superior characteristics in both convergence speed and final performance compared to conventional methods, successfully balancing optimization speed and practical usability

# 2    The EAGLE Algorithm

In this section, we detail the EAGLE algorithm. EAGLE introduces a novel update rule: **EAGLE Update Rule** that enhances parameter update efficiency, and an **Adaptive Switching Mechanism** that improves training stability.

## 2.1    EAGLE Update Rule

The foundation of EAGLE is the "EAGLE update rule," a simple approach designed to efficiently approximate and utilize the curvature information of the loss function. It is formulated as follows:

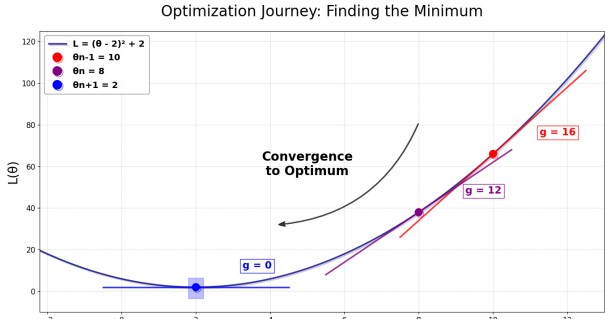

Figure 2: Visualization of the parameter update process in the example $L = (\theta - 2)^2 + 2$. Three key points are shown: $\theta_{n-1} = 10$ with $g = 16$, $\theta_n = 8$ with $g = 12$, and the optimal parameter $\theta_{n+1} = 2$ with $g = 0$.

$$\theta_{n+1} = \theta_n - \eta \cdot \frac{\theta_n - \theta_{n-1}}{\dfrac{\partial L}{\partial \theta_n} - \dfrac{\partial L}{\partial \theta_{n-1}}} \cdot \frac{\partial L}{\partial \theta_n} \tag{1}$$

The EAGLE update rule utilizes not only the current parameters $\theta_n$ and gradient $\partial L/\partial \theta_n$, but also the previous step's parameters $\theta_{n-1}$ and gradient $\partial L/\partial \theta_{n-1}$. By calculating the differences between these values and taking their ratio, we can estimate the curvature of the loss function between the current and previous steps.

The EAGLE update rule can be theoretically justified as an approximation of Newton's method. In Newton's method, the update is $\theta_{t+1} = \theta_t - H_t^{-1} g_t$, where $H_t$ is the Hessian matrix. While computing the full Hessian matrix is prohibitively expensive, it is possible to approximate $H_t^{-1}$ using the ratio of differences, providing a directional estimate of the inverse Hessian-gradient product along the optimization trajectory. Through this formulation, EAGLE implicitly captures curvature information without explicitly computing second derivatives, achieving a balance between the fast convergence and the computational efficiency.

To verify the effectiveness of the EAGLE update rule, we consider a simple quadratic function $L = (\theta - 2)^2 + 2 = \theta^2 - 4\theta + 6 \quad (\partial L/\partial \theta = 2\theta - 4)$ as an example in Fig. 2. For theoretical clarity, we omit the explicit learning rate $\eta$ from equation (1) in this example—while in practice, a learning rate improves stability in complex neural-network loss landscapes (Appendix A). Setting $\partial L/\partial \theta_n = g_n$, we apply the EAGLE update rule to estimate the optimal parameter $\theta_{n+1}$.

$$\theta_{n+1} = \theta_n - \frac{\theta_n - \theta_{n-1}}{g_n - g_{n-1}} \cdot g_n \quad \Rightarrow \quad \theta_{n+1} = 8 - \frac{8 - 10}{12 - 16} \cdot 12 = 2$$

At $\theta_{n+1} = 2$ in the original function $L = (\theta - 2)^2 + 2$, the gradient becomes $g_{n+1} = 2 \cdot 2 - 4 = 0$, coinciding exactly with the minimum point. This result demonstrates that, for a purely quadratic loss, one EAGLE update suffices to locate the minimizer. By generalizing this behavior, we expect that in neural-network training—where the loss is locally well-approximated by a downward-opening quadratic—EAGLE's update rule can significantly accelerate convergence.

## 2.2 Adaptive Switching Mechanism

The EAGLE update rule presents two primary challenges in practical applications. To address these issues, we introduce an "Adaptive Switching Mechanism" for the update rule, which aims to improve the overall stability of training. Specifically, EAGLE operates in conjunction with Adam: when conditions indicate that the EAGLE update rule is likely to be unstable or ineffective, the algorithm automatically switches to an Adam update rule. We adopt Adam as the alternative method due to its widely recognized robustness and generalization performance. However, the switching framework itself is agnostic to the choice of backup optimizer and can, in principle, accommodate any suitable

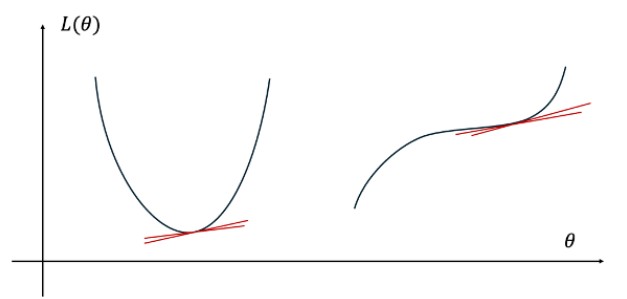

Figure 3: Gradient Change Patterns Causing Divergent Updates: Two Example Scenarios

alternative. In the following two subsections, we first describe the practical limitations of the EAGLE update rule, and then present the specific criteria under which the update rule transitions between EAGLE and Adam.

### 2.2.1 Based on Gradient Difference

The first practical challenge arises when the gradient difference $\Delta(\partial L/\partial \theta)$ becomes exceedingly small, causing the update magnitude to diverge. This phenomenon is characterized by the following relationship:

$$\Delta \frac{\partial L}{\partial \theta} \to 0 \quad \Rightarrow \quad \frac{\Delta \theta}{\Delta \frac{\partial L}{\partial \theta}} \to \pm \infty \quad \Rightarrow \quad \theta_{n+1} = \theta_n - \eta \cdot \frac{\Delta \theta}{\Delta \frac{\partial L}{\partial \theta}} \cdot \frac{\partial L}{\partial \theta_n} = \theta_n \mp \infty \to \mp \infty$$

This instability typically manifests in two common phases in Fig 3. First, during the late stages of training, gradients tend to become small as parameters approach minima. Consequently, the gradient differences between consecutive steps can become extremely small, potentially causing the divergent update (left of Fig 3). Second, in locally flat regions, gradients remain nearly constant at small values, similarly leading to extremely small gradient differences (right of Fig 3). To address this challenge, we introduce the following branching mechanism:

$$condition1 : \left| \Delta \left( \frac{\partial L}{\partial \theta} \right) \right| < threshold, \quad update\_rule = \begin{cases} Adam & \text{(if condition1)} \\ EAGLE & \text{(otherwise)} \end{cases} \tag{2}$$

This mechanism enables a flexible update strategy: Adam is applied in regions where gradient changes are minimal (to ensure numerical stability), while EAGLE is applied in regions where gradient changes are sufficiently large, taking advantage of its fast convergence properties.

The *threshold* is a critical hyperparameter in this design. Since the optimal threshold varies depending on the training stage and network architecture, a single fixed threshold cannot adequately guarantee versatility. If the threshold is set too large, the optimizer defaults too frequently to Adam, negating the benefits of EAGLE. If it is too small, the intended effect—mitigating update divergence—is lost. To overcome this, we introduce an adaptive thresholding mechanism based on the statistical properties of recent gradient norms. The threshold $\tau_n$ is dynamically computed as follows:

$$\tau_n = \min \left( \max \left( \alpha \cdot \frac{\sigma_g}{\mu_g + \epsilon}, \tau_{\min} \right), \tau_{\max} \right) \tag{3}$$

where $\mu_g$ and $\sigma_g$ denote the mean and standard deviation of the gradient norms over the last 10 steps. The ratio $\sigma_g/\mu_g$ represents the coefficient of variation (CV), which quantifies the relative variability of gradients. The hyperparameter $\alpha$ is a scaling factor (we use $5 \times 10^{-3}$ in our experiments), while $\tau_{\min}$ and $\tau_{\max}$ (set to $10^{-5}$ and $10^{-2}$, respectively) define lower and upper bounds for the threshold. $\epsilon = 10^{-8}$ is a small constant added for numerical stability. This mechanism is activated from the fifth training step; before that, a fixed initial threshold $\tau_0$ is used. When gradient variability is high

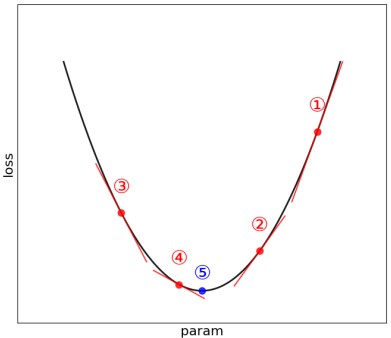
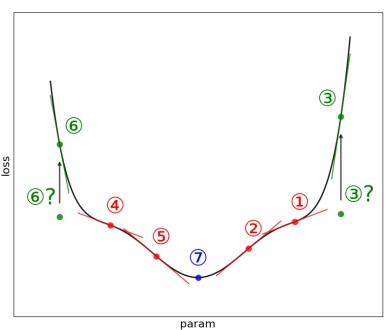

(a) Consistently Positive Second Derivative

(b) Sign-Changing Second Derivative

Point with 1: large positive gradient, 2:small positive gradient, 3:large negative gradient, 4:small negative gradient, 5:zero gradient (global minimum)

Point with 1: small positive gradient, 2:large positive gradient, 4:small negative gradient, 5:large negative gradient, 7:zero gradient (global minimum). 3,6:Incorrect estimation point

Figure 4: Illustrative Loss Landscapes

(i.e., large CV), the loss surface is assumed to be complex, and the threshold increases, favoring Adam updates. Conversely, when the CV is small, indicating smoother loss landscapes, the threshold decreases, allowing EAGLE updates to take precedence. Through this design, the optimizer can automatically switch between update rules depending on the local gradient behavior.

### 2.2.2 Based on Loss Landscape

The second practical challenge arises when the loss surface is locally non-convex. The EAGLE update rule is expected to function effectively when the loss function has no local solutions and can be approximated as downward convex. However, even if the global landscape is approximately convex, there may be local regions where the curvature is upward-opening, in which case EAGLE update can fail to achieve optimal parameter estimation. To address this challenge, we categorize the loss function shapes and its variation pattern based on the sign of the second derivative, and analyze EAGLE's effectiveness in each case.

**Consistently Positive Second Derivative**  When the loss function is globally convex—that is, its second derivative is strictly positive—the gradient changes monotonically with the parameter in the left side of Fig 4. Transitions between four key points on this curve fall into three basic patterns:

- Transition 1–1: Monotonic decrease in positive gradient (Point 1→Point 2)
- Transition 1–2: Monotonic increase in negative gradient (Point 3→Point 4)
- Transition 1–3: Oscillation near the optimum (Point 2 ↔ Point 4)

For convergence analysis, we validate using the simple quadratic function shown in Fig 2. Since the effectiveness for Transition 1–1 has already been confirmed in section 2.1, we focus on the Transitions 1–2(equation (4)) and 1–3(equation (5)).

$$(\theta_{n-1}, g_{n-1}) = (-8, -20), \ (\theta_n, g_n) = (-3, -10) \quad \Rightarrow \quad \theta_{n+1} = -3 - \frac{5}{10} \cdot (-10) = 2 \quad (4)$$

$$(\theta_{n-1}, g_{n-1}) = (-1, -6), \ (\theta_n, g_n) = (5, 6) \quad \Rightarrow \quad \theta_{n+1} = 5 - \frac{6}{12} \cdot 6 = 2 \quad (5)$$

The obtained $\theta_{n+1} = 2$ coincides with the parameter that minimizes the function. Therefore, when the loss function is globally convex and its gradient changes monotonically, the EAGLE update

Table 1: Gradient Sign Transitions and EAGLE Effectiveness

| Transition | $\nabla L_{n-1}$ | $\nabla L_n$ | $\Delta \nabla L$ | EAGLE Effective? |
|:---:|:---:|:---:|:---:|:---:|
| 1–1 | Large + | Small + | − | ✓ |
| 1–2 | Large − | Small − | + | ✓ |
| 1–3 | +/− | −/+ | −/+ | ✓ |
| 2–1 | Small + | Large + | + | ✗ |
| 2–2 | Small − | Large − | − | ✗ |

rule exhibits convergence toward the global minimum across all fundamental transition patterns, confirming its effectiveness.

**Sign-Changing Second Derivative** When the loss surface exhibits local concave–convex structure (i.e., the second derivative changes sign), the gradient response to parameter changes becomes more complex in the right side of Fig 4. In this scenario, two new transition patterns not observed in the "consistently positive second derivative" case exist:

- Transition 2–1: Increasing positive gradient (Point 1→Point 2)
- Transition 2–2: Decreasing negative gradient (Point 4→Point 5)

Since the EAGLE update rule is designed to drive the gradient toward zero under the assumption of downward curvature, applying it here can lead to incorrect updates that increase the loss. Specifically in Fig 4, in Transition 2-1, Point 3?, and in Transition 2-2, Point 6? are estimated as parameters where the gradient converges to zero, and the actual losses and gradients at these parameters are at Points 3 and 6, indicating inappropriate updates. We summarize these findings by examining the signs of the previous gradient $\nabla L_{n-1}$, current gradient $\nabla L_n$, and gradient change $\Delta \nabla L$ in Table 1. From this, we identify two simultaneous conditions under which the EAGLE update rule fails: "Consistency of gradient signs between consecutive steps" and "Consistency of signs between current gradient and gradient change". To handle these inappropriate updates, we introduce the following branching mechanism:

$$condition2 : (\nabla L_{n-1} \cdot \nabla L_n \geq 0) \wedge (L_n \cdot \Delta \nabla L \geq 0), \quad update\_rule = \begin{cases} Adam & (if\ condition2) \\ EAGLE & (otherwise) \end{cases}$$
(6)

### 2.2.3 Integration

Based on these analyses, we integrate the two branching criteria into a unified switching mechanism. Both the gradient-difference threshold check from section 2.2.1 (condition 1) and the loss-landscape check from section 2.2.2 (condition 2) serve to trigger a switch to the Adam update. Therefore, we combine them with a logical OR and formulate a single selection rule:

$$condition : condition1 \vee condition2, \quad update\_rule = \begin{cases} Adam & (if\ condition) \\ EAGLE & (otherwise) \end{cases}$$
(7)

This unified branching mechanism enables a consistent handling of both practical challenges—namely, divergence caused by vanishing gradient differences and incorrect updates in locally concave regions—within a single framework.

## 3 Related Work

**Adaptive Moment Estimation: Adam** is an optimization method that combines the advantages of AdaDelta [15] , Momentum [16], and RMSprop [17]. The first-order moment (mean) incorporates the Momentum concept similar to AdaDelta, calculating the moving average of gradients to consider the directional trends of past gradients and appropriately determine the update direction. This

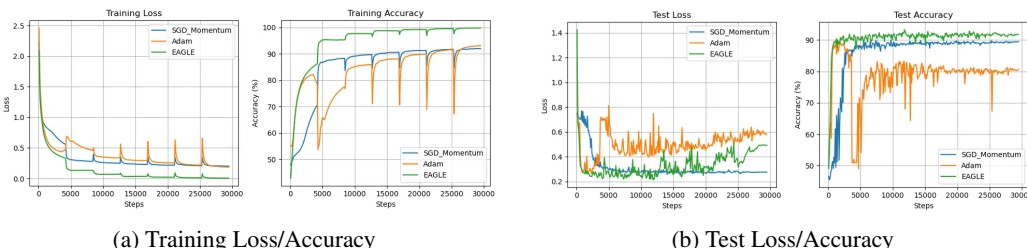

(a) Training Loss/Accuracy                    (b) Test Loss/Accuracy

Figure 5: Results on SST-2 (GPT-2) Text Classification

Table 2: Optimizer Training/Test Loss and Accuracy on GPT-2 (SST-2) Task

| Optimizer | Train Loss | Train Acc (%) | Test Loss | Test Acc (%) |
|---|---|---|---|---|
| SGD_Momentum | 0.2035 | 91.98 | **0.2760** | 89.45 |
| Adam | 0.1863 | 93.15 | 0.5773 | 80.62 |
| EAGLE | **0.0057** | **99.81** | 0.4929 | **91.74** |

Table 3: EAGLE Update Usage Analysis on GPT-2 (SST-2) Task

| Optimizer | Avg Usage (%) | Min Usage (%) | Max Usage (%) |
|---|---|---|---|
| EAGLE | 20.50 | 12.55 | 32.11 |

mechanism enables stable updates even with significant gradient changes. The second-order moment
(variance) incorporates the RMSprop concept similar to AdaDelta, calculating the moving average
of squared gradient values to adaptively adjust learning rates individually for each parameter. This
mechanism enables efficient learning based on parameter importance and update necessity. While
Adam has relatively many hyperparameters, they exhibit high robustness, and their default values
($\beta_1 = 0.9, \beta_2 = 0.999, \epsilon = 1e - 8$) function well across a wide range of tasks.

# 4 Experiments

In this study, we evaluate SGD with Momentum, Adam, and our proposed EAGLE on two tasks.
The first is text classification—fine-tuning GPT-2 Small (117 M parameters) on SST-2 (GLUE). The
second is image classification—fine-tuning ViT-B/16 (86 M parameters) on CIFAR-10. We record
convergence speed, training/testing loss and accuracy, and the usage ratio change in EAGLE update
rule as evaluation metrics. Experimental setup is detailed in Appendix B.

## 4.1 Text Classification

Figure 5 and Table 2, 3 summarize the fine-tuning results of GPT-2 Small on SST-2.

**Convergence Speed**   We measure how many steps EAGLE requires to reach the final training
loss achieved by each baseline optimizer. Out of a total of 29,469 steps, EAGLE reaches SGD
with momentum's final loss (0.2035) in 4,313 steps—an approximately 6.83× speedup. Similarly, it
reaches Adam's final loss (0.1863) in 4,355 steps—an approximately 6.77× speedup. These results
demonstrate that EAGLE substantially accelerates loss convergence on the SST-2 text classification
task, fulfilling the primary objective of this study.

**Final Performance**   For final training loss, EAGLE achieved values 98.06% and 96.94% lower than
SGD with momentum and Adam, respectively. For final training accuracy, EAGLE outperformed
them by 7.83% and 6.66%. For final test accuracy, EAGLE outperformed them by 2.29% and 11.12%.
Furthermore, EAGLE consistently outperformed the other optimizers in test accuracy throughout most
of the training process. These results indicate that EAGLE not only achieves stable learning, fulfilling
the primary objective of this study, but also contributes to improving final model performance.

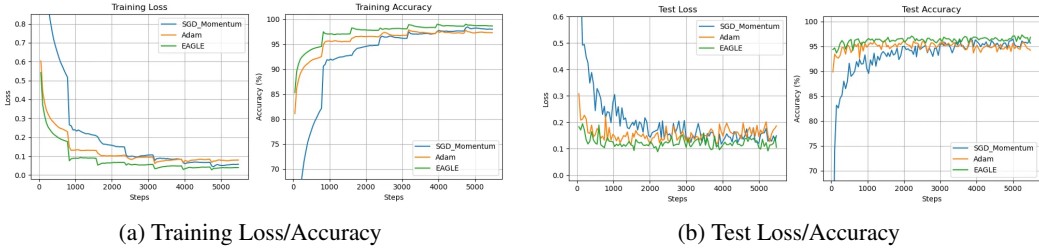

(a) Training Loss/Accuracy        (b) Test Loss/Accuracy

Figure 6: Results on CIFAR-10 (ViT-B/16) Image Classification

Table 4: Optimizer Training/Test Loss and Accuracy on ViT-B/16 (CIFAR-10) Task

| Optimizer | Train Loss | Train Acc (%) | Test Loss | Test Acc (%) |
|---|---|---|---|---|
| SGD_Momentum | 0.0574 | 98.03 | 0.1493 | 95.67 |
| Adam | 0.0797 | 97.33 | 0.1865 | 94.22 |
| EAGLE | **0.0396** | **98.68** | **0.1059** | **96.84** |

Table 5: EAGLE Update Usage Analysis on ViT-B/16 (CIFAR-10) Task

| Optimizer | Avg Usage (%) | Min Usage (%) | Max Usage (%) |
|---|---|---|---|
| EAGLE | 9.54 | 7.62 | 14.78 |

**EAGLE Update Usage Ratio** The EAGLE update usage ratio shows a roughly monotonic decrease as training progresses (Appendix C). This indicates that the EAGLE update rule is actively applied in the high-loss, high-gradient regime of early training, then transitions to the more stable Adam update rule as convergence approaches. Therefore, we can conclude that the adaptive branching mechanism functions effectively, successfully balancing efficient exploration and stable convergence.

## 4.2 Image Classification

Figure 6 and Table 4, 5 summarize the fine-tuning results of ViT-B/16 on the CIFAR-10.

**Convergence Speed** Out of a total of 5,473 training steps, EAGLE reaches SGD with momentum's final loss (0.0574) in 1,607 steps—an approximately 3.41× speedup. Likewise, it reaches Adam's final loss (0.0797) in 829 steps—an approximately 6.60× speedup. These results demonstrate that EAGLE successfully accelerates loss convergence on the image classification task as well, fulfilling our primary objective.

**Final Performance** For final training loss, EAGLE achieved values 31.01% and 50.31% lower than SGD with momentum and Adam, respectively. For final training accuracy, EAGLE outperformed them by 0.65% and 1.35%. For final test accuracy, EAGLE exceeded them by 1.17% and 2.62%. Notably, EAGLE already reaches the baselines' final test accuracy levels in the early phase and then consistently outperforms them thereafter. These results indicate that EAGLE not only achieves stable learning but also contributes to improving final model performance.

**EAGLE Update Usage Ratio** While the EAGLE update rule usage ratio is generally lower compared to the text classification task (Appendix C), considering its strong convergence speed and final performance, it remains effective in the image recognition domain.

Based on these results, the proposed EAGLE optimizer outperforms the representative baselines—SGD with Momentum and Adam—on both GPT-based text classification and ViT-based image classification in terms of convergence speed and final performance, thereby achieving the objectives of this study.

Table 6: Time Efficiency (Seconds per Epoch)

| Optimizer | GPT Model | | ViT Model | |
|---|---|---|---|---|
| | Python | PyTorch | Python | PyTorch |
| SGD_Momentum | 160.39s | 110.70s | 191.04s | 19.80s |
| Adam | 361.06s | 110.19s | 348.53s | 22.05s |
| EAGLE | 897.06s | - | 447.33s | - |

## 5 Limitations

While EAGLE demonstrates impressive convergence acceleration and performance improvements in our experiments, we acknowledge several limitations of our current approach.

**Is Adam Really The Best Choice?** Our EAGLE algorithm was implemented with an adaptive switching mechanism that by default uses Adam as the backup optimizer (due to its known robustness). We also implemented a variant using SGD with momentum as the fallback; however, the Adam-based configuration consistently yielded better training outcomes (Appendix D).

**Tradeoffs In Steps vs Wall-Clock Time** In our pure-Python implementation, EAGLE requires approximately 1.29–5.59× more wall-clock time per epoch compared to SGD with momentum or Adam (Tab 6). Furthermore, because the PyTorch/C++ backend for EAGLE has not yet been developed, the gap widens when comparing against PyTorch-optimized SGD with momentum and Adam. Consequently, the per-step runtime overhead negates EAGLE's advantage in step-based convergence speed.

## 6 Conclusion and Future works

In this paper, we proposed EAGLE, an optimizer that combines lightweight curvature-aware update rule with an adaptive switching mechanism to Adam. In experiments on both an NLP task (fine-tuning GPT-2 on SST-2) and a vision task (fine-tuning ViT-B/16 on CIFAR-10), EAGLE converged faster than standard optimizers – roughly 6.83× faster than SGD with momentum and 6.77× faster than Adam on SST-2, and 3.41× faster than SGD and 6.60× faster than Adam on CIFAR-10– while also achieving better final accuracy in both cases. These results indicate that EAGLE effectively accelerates loss convergence and often leads to higher generalization performance. In summary, EAGLE leverages simple curvature estimates from gradient differences to take large early steps, then smoothly falls back to Adam when needed for stability, successfully combining fast exploration with stable convergence.

Future work will pursue several directions to make EAGLE more practical and broadly applicable:

1. **Optimized implementation:** Develop a PyTorch/C++ (GPU) version of EAGLE to eliminate the current Python bottleneck and achieve per-step speed comparable to other optimizers.

2. **Broader applications:** Apply EAGLE to other domains such as generative modeling (e.g. GANs or diffusion models) and reinforcement learning, to test its benefits beyond supervised tasks.

3. **Theoretical analysis:** Investigate the convergence properties and switching behavior of EAGLE formally, to better understand when and why the adaptive strategy yields gains.

These directions aim to address the current limitations and unlock the full potential of curvature-aware adaptive optimization for future deep learning applications.

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

# Appendix

## A  Effect of EAGLE's Learning-Rate Adaptation

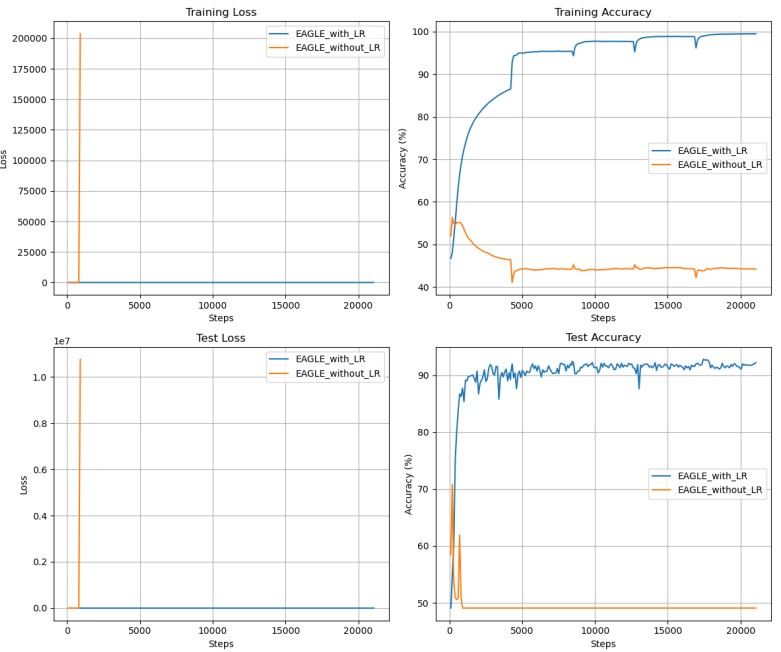

Figure 7: GPT-2 on SST-2

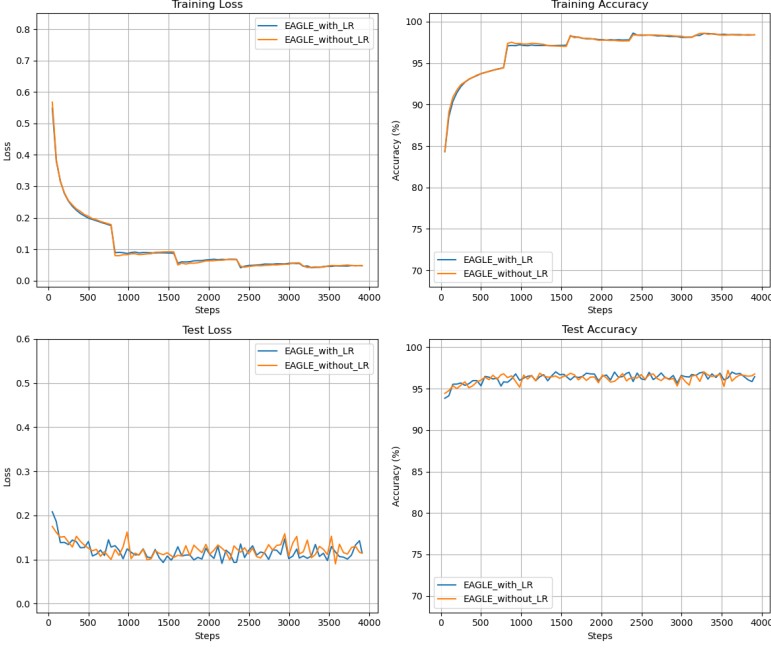

Figure 8: ViT-B/16 on CIFAR-10

# B   Experimental Setup

All experiments were conducted on a single NVIDIA A100 80 GB PCIe GPU.

Table 7: Experimental settings for evaluating optimization algorithms across different tasks.

| Task | Hyperparameters | | |
| --- | --- | --- | --- |
| | SGD w/ Momentum | Adam | EAGLE (Ours) |
| Text Classification | lr = 5e-5
momentum = 0.9
weight decay = 1e-2
batch size = 16 | lr = 5e-5
$\beta_1 = 0.9$, $\beta_2 = 0.999$
weight decay = 1e-2
batch size = 16 | lr = 5e-5
threshold = 1e-4
weight decay = 1e-2
batch size = 16 |
| Image Classification | lr = 1e-2
momentum = 0.9
weight decay = 1e-4
batch size = 64 | lr = 1e-4
$\beta_1 = 0.9$, $\beta_2 = 0.999$
weight decay = 1e-4
batch size = 64 | lr = 1e-4
threshold = 5e-4
weight decay = 1e-4
batch size = 64 |

# C   Detailed Changes of EAGLE Update Rule Usage Ratio

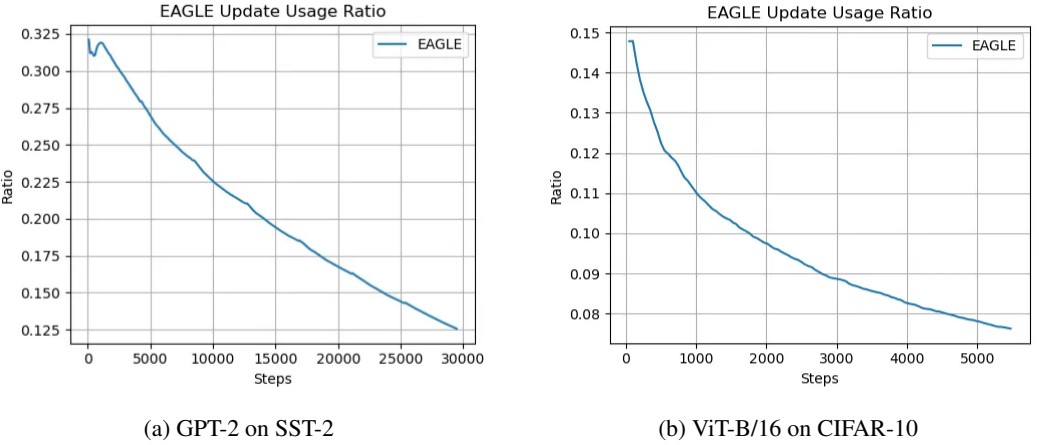

(a) GPT-2 on SST-2                    (b) ViT-B/16 on CIFAR-10

Figure 9: EAGLE Update Rule Usage Ratio

## D    Adam-based EAGLE (EAGLE, EAGLE-A) vs SGD with Momentum-based EAGLE (EAGLE-S)

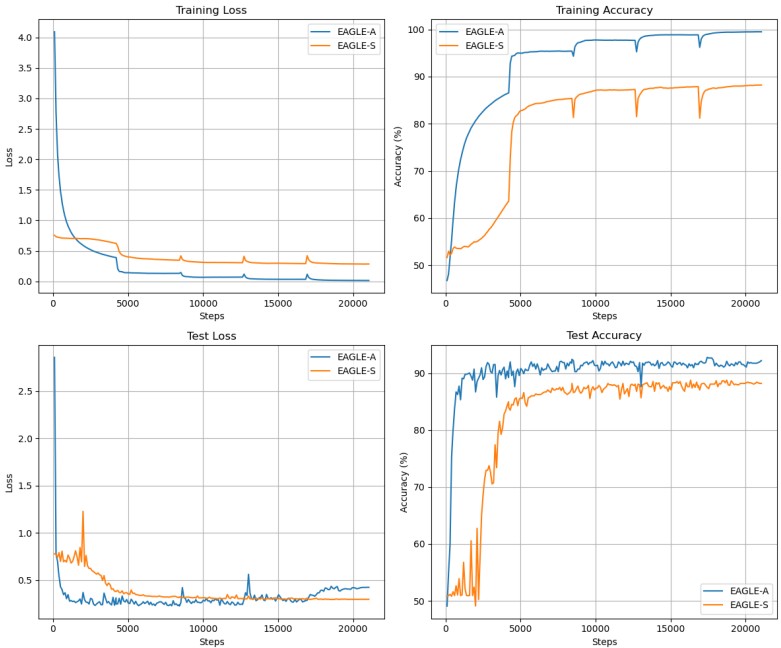

Figure 10: GPT-2 on SST-2

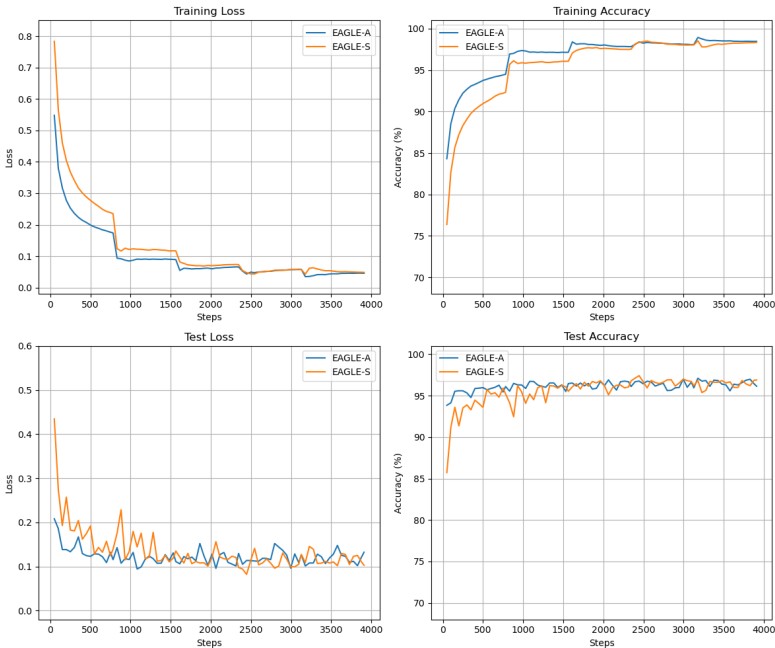

Figure 11: ViT-B/16 on CIFAR-10

