# OpenReview forum: "EAGLE: Early Approximated Gradient-based Learning-rate Estimator"
_NeurIPS.cc/2025/Conference — Submitted to NeurIPS 2025_

### Official Review · Reviewer_hHtm · 2025-06-16

**Clarity:** 2
**Significance:** 2
**Originality:** 1
**Rating:** 2
**Confidence:** 3

**Summary:**

EAGLE is studied as a preconditioner for single-node optimization. In the context of deep neural network (DNN) optimization, computing the Hessian matrix and its inverse introduces significant computational overhead. As a result, lightweight methods like Adam, which adaptively adjust the learning rate, are commonly preferred. EAGLE, as formulated in Eq. (1), achieves adaptive learning rate adjustment by utilizing update differences in parameters and gradients. Additionally, it demonstrates stable behavior by switching to Adam in response to the scenarios of the loss landscape. Through two DNN benchmark tasks, EAGLE consistently outperforms Adam in both convergence speed and final performance.

**Questions:**

None in particular.

**Ethical Concerns:**

["NO or VERY MINOR ethics concerns only"]

**Limitations:**

The limitations of the empirically conducted experiments are discussed in Section 5; however, the social impact of the proposed method is not addressed, either in the main text or in the Appendix.

**Paper Formatting Concerns:**

The GitHub link allows identification of the authors, which also appears to be a major issue. I strongly recommend anonymizing the submission.

**Quality:**

1

**Strengths And Weaknesses:**

__Strengths__

- (S1) The authors provide an intuitive explanation of EAGLE’s behavior using illustrations based on the loss landscape. In addition, they propose a mechanism that switches to Adam to mitigate instability in convergence.

- (S2) The experimental results consistently demonstrate that EAGLE outperforms Adam across two benchmark tests.

- (S3) The source code is publicly available, allowing readers to easily try out the method.

__Weaknesses__
- (W1) Insufficient Motivation for Introducing the Update Rule

Line 68 states: “EAGLE update rule can be theoretically justified as an approximation of Newton’s method.” However, does the preconditioning matrix used in Eq. (1) actually approximate the inverse Hessian $H^{-1}$?

Although the example presented from line 75 attempts to explain the motivation for the update rule, it feels more like an empirical discussion. From a reviewer’s perspective, this is insufficient as a principled motivation for the preconditioner formulation.

- (W2) Lack of Convergence Analysis

Even if limited to convex functions, a theoretical convergence analysis would be valuable.

The switching mechanism between Adam and EAGLE is currently based on empirical heuristics. How does this switching behavior affect the convergence rate?

- (W3) Small-Scale Experiments in Section 4

While the comparisons between SGDm, Adam, and EAGLE are informative, many other optimizers have been proposed in this field. The scope of optimizer comparison is too limited.

The choice of benchmark tasks (CV and NLP) seems reasonable, but if the authors wish to emphasize empirical evidence, larger-scale benchmark evaluations would strengthen the argument.

- (W4) Lack of Runtime Comparison on PyTorch

- Since Eq. (1) appears to use update differences of parameters and gradients, implementing it on PyTorch and comparing the actual runtime should be feasible. (It is anticipated that EAGLE is unlikely to be critically slow.)

---

> ### Author Rebuttal · Authors · 2025-07-28
>
> Thank you for your time and for providing valuable feedback that will help us improve the clarity and rigor of our paper.
>
> - **W1: On Insufficient Motivation:**
> We agree that the motivation in the initial draft was not sufficiently principled. We will revise the paper to reframe our method as a "stabilized secant method," grounding it in classical numerical analysis. The novelty will be clearly articulated as the adaptive switching mechanism that makes this method practical for deep learning.
>
> - **W2: On Lack of Convergence Analysis:**
> This is a valid point. A full convergence proof for a hybrid optimizer is a challenging task. However, by reframing the method based on the secant method, we can better connect it to existing convergence literature. We will add this as a key direction for future theoretical work in our conclusion.
>
> - **W3: On Small-Scale Experiments:**
> We agree that expanding our experiments to include more SOTA optimizers and more challenging tasks (e.g., training from scratch) is essential. While this is not feasible during the rebuttal period, we acknowledge this limitation and have made it a priority for our future research.
>
> - **W4: On Lack of Runtime Comparison:**
> Thank you for this suggestion. We would like to clarify that this analysis is already present in the manuscript in Section 5 (Limitations) and Table 6. These sections discuss the wall-clock time overhead of our current Python implementation and identify an optimized C++/GPU version as critical future work. We will ensure this point is more clearly signposted in the revision.

---

> > ### Comment · Reviewer_hHtm · 2025-08-04
> >
> > Thank you for your response. However, I cannot say that my concerns have been fully addressed. I keep my original score. I believe the effectiveness of the algorithm would be better supported through theoretical convergence analysis and/or more extensive empirical validations.

---

### Official Review · Reviewer_wBFS · 2025-06-28

**Clarity:** 3
**Significance:** 2
**Originality:** 1
**Rating:** 2
**Confidence:** 4

**Summary:**

The paper introduces the *Early Approximated Gradient based Learning-rate Estimator* (EAGLE) optimisation algorithm, which adaptively switches between ADAM and a secant-method-type update rule. The authors evaluate EAGLE on language and vision fine-tuning tasks.

**Questions:**

1. How sensitive are EAGLEs hyper-parameters? Given EAGLE adds additional knobs to turn, and can reduce to ADAM in case they are chosen large, the cost of hyper-parameter tuning should be discussed.
2. How large is the memory overhead of EAGLE compared to Adam?
3. How does the optimiser perform compared to other curvature-aware optimisers such as LION [9] or MUON [10, 11]? Even small-scale comparisons would help to position the contribution more clearly.

---

### References

[9] Chen et al., Symbolic Discovery of Optimization Algorithms, NeurIPS 2023

[10] Jordan et al., Muon: An optimizer for hidden layers in neural networks, Blog post, 2024

[11] Shen et al, On the Convergence Analysis of Muon, arXiv 2025

**Ethical Concerns:**

["NO or VERY MINOR ethics concerns only"]

**Final Justification:**

During the rebuttal, the authors agreed that multiple changes are required, including

- positioning the work as a *stabilized secant method*, instead of a novel method, and

- adding empirical comparisons to other curvature-aware optimisers.

These changes require a major revision of the work and its positioning within the related literature, and hence a new review after these changes are implemented. In particular I recommend to reject the work at this stage, but recommend the authors to implement the mentioned changes and re-submit the work at a future venue.

**Limitations:**

Yes

**Paper Formatting Concerns:**

The abstract contains a github repository with a revealing github username, breaking the double-blind review requirement.

**Quality:**

2

**Strengths And Weaknesses:**

## Strengths
- The paper addresses the important question of adaptive step-size selection in machine learning tasks.
- The algorithm description is clear, and the overall paper is well-written and easy to follow.

## Weaknesses
The primary weakness of this work is its limited novelty. The proposed EAGLE update rule is the component-wise secant method
$$
\theta_{n+1} \gets \theta_n - \eta \frac{\theta_n - \theta_{n-1}}{\nabla L (\theta_n) - \nabla L(\theta_{n-1})} \odot \nabla L (\theta_n),
$$
which in its core has been known since 1800 BC [1], and has been around in its modern form since the 16th century [2]. Hence the core method itself lacks novelty, and key historical and modern references are missing.
Furthermore, adaptive switching between different optimization rules have also been extensively studied in the last century [3-5], and more recently in [6] and references therein.  This literature is not discussed in the current submission. Although the specific adaptive conditions employed in this work appear novel, the sign-based condition closely resembles known methods such as Brent’s algorithm [7].

Additionally, the experimental setup has methodological weaknesses. The hyper-parameter tuning process is not transparently described, making it unclear whether the baseline methods were tuned. Moreover, the provided implementation uses AdamW for EAGLE within its adaptive switching mechanism, whereas the baseline utilizes Adam with L2 regularization. Given that AdamW is known to outperform Adam with L2 regularization [8], this discrepancy favours the proposed method.

Finally, the work does not provide any theoretical guarantees for the proposed algorithm.

Below a list of possible typos.
- Line 18: Intrduction -> Introduction
- Line 235: Tab -> Table

---

### References

[1] Papakonstantinou and Tapia, Origin and Evolution of the Secant Method in One Dimension, Amer. Math. Monthly, 2013.

[2] Cardano, Artis magnae, sive de regulis algebraicis (Ars magna), Nuremberg, 1545.

[3] Wu et al., Google’s neural machine translation system: Bridging the gap between human and machine translation, arXiv, 2016.

[4] Akiba et al., Extremely large minibatch SGD: Training resnet-50 on ImageNet in 15 minutes, arXiv, 2017.

[5] Keskar and Socher, Improving Generalization Performance by Switching from Adam to SGD, arXiv, 2017.

[6] Schröder et al., Switching between Numerical Black-box Optimization Algorithms with Warm-starting Policies, arXiv, 2022.

[7] Brent, Algorithms for minimization without derivatives, Courier Corporation, 2013.

[8] Loshchilov and Hutter, Decoupled Weight Decay Regularization, ICLR 2019

---

> ### Author Rebuttal · Authors · 2025-07-28
>
> Thank you for your detailed review and for providing numerous valuable suggestions to improve our work. We appreciate your historical perspective and your sharp eye for methodological detail. We address your points below.
>
> - **On Limited Novelty (Weakness 1)**
> We thank you for your historical insight and agree that the core update rule is indeed the secant method. In our revision, we will embrace this connection, explicitly citing the relevant historical and modern literature. We will clarify that our claim to novelty is not in the invention of the update rule itself. Rather, the novelty lies in the fact that applying this classical method to optimize neural networks with billions of parameters is a non-trivial endeavor. The core contribution of our work is the design of the adaptive switching framework that makes this classical, but often unstable, method practical and effective in the complex domain of modern deep learning. This stabilization framework is what distinguishes EAGLE from prior work.
>
> - **On Methodological Flaws (Weakness 2)**
> You have identified a significant methodological flaw in our experiments, and we sincerely apologize for this oversight. Thank you for bringing it to our attention. In the revised manuscript, we will make the following corrections to ensure a fair and transparent comparison:
>
>   1. **Fair Comparison:** We will replace the Adam baseline with AdamW in all experiments. This ensures a direct and fair comparison, as EAGLE's fallback mechanism is indeed more akin to AdamW.
>
>   2. **Transparency:** We will add a new section to the appendix detailing the hyperparameter tuning protocol for all optimizers, including the search space and selection criteria, to ensure full reproducibility.
>
> - **On Theoretical Guarantees and Practical Questions (Weakness 3, Questions)**
> This is a valid point. A full convergence proof for a hybrid, non-convex optimizer like EAGLE is a highly challenging task that is likely beyond the scope of a single empirical paper. However, by reframing our method as a "stabilized secant method," we strengthen its connection to existing convergence literature. We will add a discussion of this to our "Future Work" section, highlighting it as an important direction for theoretical research.
>
> - **Responses to your questions:**
>
>   - Hyperparameter Sensitivity: This is an excellent question. We will add a new ablation study to the appendix of the revised paper to analyze the sensitivity of EAGLE's key hyperparameters.
>
>   - Memory Overhead: We will clarify in the main text that EAGLE's memory overhead is O(N), identical to Adam, as it only requires storing the parameters and gradients from the previous step. The computational overhead is already discussed in Section 5 and Table 6 of our paper.
>
>    - Comparison to LION/MUON: We agree that comparing against modern optimizers like LION  and MUON  is important for positioning our work. While conducting these experiments is not feasible during the rebuttal period, we will add a qualitative comparison in our Related Work section discussing their different design principles (LION: sign-based momentum; MUON: matrix orthogonalization) and will mark experimental comparison as important future work.

---

> > ### Comment · Reviewer_wBFS · 2025-08-02
> >
> > I sincerely appreciate the authors' openness and constructive attitude towards the feedback provided by the reviewers.
> >
> > Given the magnitude and significance of the required revisions, along with the authors' acknowledgment that these cannot all be completed within the rebuttal period, I am inclined to maintain my current score. In particular, the primary areas that still require improvements include the discussion of related work, as well as either the more extensive empirical validation of the algorithm or, alternatively, theoretical guarantees.
> > Nevertheless, I recognize the potential value of the proposed algorithm and encourage the authors to implement the suggested revisions and resubmit their improved manuscript to a future venue.

---

### Official Review · Reviewer_h7iv · 2025-07-02

**Clarity:** 2
**Significance:** 1
**Originality:** 2
**Rating:** 2
**Confidence:** 3

**Summary:**

This paper introduces EAGLE (Early Approximated Gradient-based Learning-rate Estimator), a novel optimization algorithm designed to accelerate training in deep learning models by incorporating lightweight curvature estimates from gradient differences. EAGLE proposes a new update rule that approximates second-order information without explicit Hessian computation and integrates an adaptive switching mechanism to fall back on Adam updates in unstable regimes. The method is evaluated on text classification (SST-2 with GPT-2) and image classification (CIFAR-10 with ViT), demonstrating significantly faster convergence and improved final accuracy compared to SGD with momentum and Adam.

**Questions:**

see Weakness

**Ethical Concerns:**

["NO or VERY MINOR ethics concerns only"]

**Final Justification:**

This version needs more improvement.

**Limitations:**

yes

**Quality:**

2

**Strengths And Weaknesses:**

**Strengths**:

- The paper considers an important problem in optimization—accelerating convergence via curvature-aware updates without explicit Hessian computation. The proposed method is conceptually simple and easy to implement, with a clear mechanism for fallback stability via integration with Adam.

**Weakness**:
- The motivation behind this work is not clearly proposed. Although the authors claim to focus on estimating second-order curvature information without explicitly computing the Hessian, the rationale for the proposed method is neither intuitively explained nor theoretically grounded. I can understand the intuition that motivates the EAGLE update rule: since the Hessian $H$ is defined as $H = \frac{\partial g}{\partial \theta}$ , then in the limit $\Delta \theta \to 0$, the ratio $\frac{\Delta g}{\Delta \theta}$ can approximate the Hessian. Consequently, the authors adopt $\frac{\Delta \theta}{\Delta g}$ as a proxy for $H^{-1}$. However, the paper lacks any discussion that would clarify or justify this approximation from either a mathematical or statistical perspective. The idea is merely presented through a toy quadratic example without elaborating on its limitations or applicability beyond this setting.
Specifically, I consider this form of curvature approximation to be relatively trivial. Crucially, the authors do not provide any analysis showing that this estimate has desirable statistical properties—e.g., whether it correlates with the true Hessian (or its inverse) in expectation or higher moments. In contrast, recent works such as Sophia [1] have proposed more principled approaches to curvature estimation using only first-order information. These methods introduce well-defined estimators for diagonal Hessian elements or Gauss-Newton approximations, and rigorously demonstrate that such estimators are unbiased or consistent under certain conditions. Further, they provide convergence guarantees—at least in the convex case—supporting the soundness of their methods. In comparison, EAGLE lacks all of these elements. The paper provides neither theoretical insight nor statistical justification for its update rule beyond a single-step illustration on a quadratic function. There is no analysis even within this simple quadratic setting (e.g., convergence upper bounds, stability regions), let alone any discussion of how the method behaves under the far more complex and non-convex landscapes characteristic of neural network training. As a result, I find it hard to be convinced that the proposed update rule offers a meaningful improvement over existing preconditioning or adaptive optimization methods. Without a clearer motivation, stronger justification, and a connection to existing literature on curvature estimation, the value of this contribution remains doubtable.

- The authors propose EAGLE as a curvature-aware optimizer that conditionally falls back to Adam when the gradient difference $\Delta g$ is below a threshold, in order to avoid instability due to potentially unbounded update magnitudes. While this adaptive switching mechanism is a reasonable design choice for ensuring training stability, the paper lacks any substantial analysis on how often and under what conditions this fallback is actually triggered during training. In particular, it is well known that during the pretraining or early fine-tuning phases of large models, the gradient norm tends to decrease rapidly—especially in the initial loss-drop phase—and then oscillates around a small value with low variance as the model gradually converges. In such stages, the difference between consecutive gradients $\Delta g$ is typically very small. As a result, I suspect that the proposed method may align with Adam for the majority of training steps, with the EAGLE update rule being applied only in the very early stages where gradients are still large and varying.
This concern is partially supported by the loss curves reported in the paper: the major divergence between EAGLE and baseline optimizers (SGD, Adam) occurs in the first few thousand steps, after which the curves follow similar trajectories. This suggests that the benefit of EAGLE might stem primarily from selecting larger effective step sizes during the early phase, rather than maintaining a consistently different or curvature-aware update trajectory throughout training. Given this, I believe the paper would strongly benefit from a detailed analysis of the dynamics of the update rule selection. Without such analysis, it is reasonable to interpret EAGLE as an ad-hoc adjustment of the learning rate schedule in the early stage, which then defaults to Adam-like behavior for the majority of training. This significantly weakens the argument that EAGLE provides a fundamentally new optimization behavior.

- The experimental results presented in the paper are not sufficiently convincing to establish the practical effectiveness of the proposed method. All evaluations are conducted solely in the context of fine-tuning pre-trained models. However, fine-tuning tasks are often relatively easy optimization problems, where convergence is less challenging and many simple methods—even some zeroth-order techniques—can yield competitive performance. As such, the ability to accelerate convergence in this context does not necessarily imply that the proposed optimizer is generally effective or robust. To truly demonstrate the validity, even training a smaller architecture like GPT-2 from scratch on a mid-sized corpus would be informative. This kind of experiment would reveal whether the curvature approximation strategy employed by EAGLE can consistently support convergence over a long training horizon and under diverse gradient regimes. If the method fails to converge in this setting—or proves to be unstable—its utility would be severely limited.
Furthermore, given that EAGLE introduces non-negligible additional computation per step (as acknowledged in the paper),in fine-tuning, where a wide variety of optimizers already perform well with minimal tuning, the computational overhead of EAGLE is hard to justify unless its advantages are clearly established in more general or challenging settings.
[1] Hong Liu, Zhiyuan Li, David Hall, Percy Liang, Tengyu Ma, Sophia: A Scalable Stochastic Second-order Optimizer for Language Model Pre-training



Given the concerns outlined above regarding the lack of theoretical justification, insufficient analysis of the update mechanism’s behavior, and limited experimental validation, I believe this paper requires substantial revisions before it can meet the standards expected for publication at NeurIPS.

---

> ### Author Rebuttal · Authors · 2025-07-28
>
> Thank you for your deeply insightful and detailed critique. Your feedback has been invaluable in identifying and strengthening the core narrative of our work.
>
> - **On Theoretical and Motivational Grounding:**
> We acknowledge that our original framing of EAGLE as an "approximate Newton method" was imprecise. Based on your feedback, we will revise our motivation to frame EAGLE as a **"stabilized secant method."** This grounds our work in classical numerical analysis  and shifts the core novelty to the **adaptive switching mechanism**, which is designed to make the classical (but unstable) secant method practical for high-dimensional, non-convex optimization. This distinguishes our pragmatic approach from principled Hessian estimators like Sophia.
>
> - **On the Analysis of Switching Dynamics:**
> This is a sharp observation. We will clarify in the revision that the hybrid nature of EAGLE is by design. The higher usage of the EAGLE update in the chaotic early training phase acts as an "ignition booster" for faster initial convergence . The subsequent switch to the more robust Adam update in later stages is an intended feature for stable fine-tuning, and the decreasing usage ratio is evidence that this mechanism is working as expected.
>
> - **On Limited Experimental Validation:**
> We fully agree that demonstrating performance on from-scratch training (e.g., on GPT-2) is a critical test of our method's general applicability. While this is not feasible to conduct during the rebuttal period, we have added it to our roadmap as a high-priority experiment for future work.

---

### Official Review · Reviewer_aHCb · 2025-07-03

**Clarity:** 1
**Significance:** 1
**Originality:** 2
**Rating:** 2
**Confidence:** 5

**Summary:**

This paper proposes a novel optimization method that accelerates loss convergence during early stages of training.

**Questions:**

1. Would it be possible if authors could provide a comparison with state-of-art works mentioned in the Weakness? I would be raise my score if EAGLE works better than those.
2. Could the authors provide more training experiments on larger models eg. Llama350M, Llama1b? To test whether this method have a boarder utility.
3. Could the authors clarify the difference between EAGLE work and existing optimization works eg. AdaQN, AdaHessian. It would lack of novelty if EAGLE method is mainly about specific threshold formula, which constitutes only an incremental improvement.

**Ethical Concerns:**

["NO or VERY MINOR ethics concerns only"]

**Final Justification:**

Thank you for your response. Taking into account the feedback from the other reviewers, and limited experiments conducted in this work, I’ve decided to keep my score unchanged.

I hope the authors will further conduct more experiments in the future to evaluate the generalizability of your proposed method. looking forward to seeing your future work.

**Limitations:**

yes

**Paper Formatting Concerns:**

The submission appears to be well-formatted according to the NeurIPS guidelines.

**Quality:**

2

**Strengths And Weaknesses:**

Strengths:
1. A novel update rule utilizing the local curvature of the loss function between consecutive steps.
2. The authors provides an adaptive switching mechanism, switching update rules according to different constraints.

Weakness:
1. The EAGLE update rules was only tested in small benchmark scale.
2. The EAGLE method didn't compare with existing methods eg. AdamW, Lion, Sophia, etc.
3. Limited Novelty. Curvature from gradient differences and hybrid optimizers have prior works eg. AdaQN, Shampoo+momentum, Sophia, AdaHessian methods, the paper does not differentiate clearly from these.

---

> ### Author Rebuttal · Authors · 2025-07-28
>
> Thank you for your constructive feedback, which has been very helpful in clarifying our paper's contributions.
>
> **On Novelty and Differentiation from Existing Work:**
> We appreciate you pushing for this clarification. Our primary novelty is not a specific threshold formula, but the framework of a stabilized secant method for deep learning. The key differentiators are:
>
> - **vs. AdaQN:** EAGLE is memoryless beyond the previous step and does not require maintaining an L-BFGS buffer.
>
> - **vs. AdaHessian:** EAGLE avoids the costly second backward pass (Hessian-vector product) required by AdaHessian , using only first-order information.
>
> We will revise the manuscript to make these distinctions explicit.
>
> **On SOTA Comparisons and Larger-Scale Experiments:**
> We agree that comparing against SOTA optimizers (like AdamW, Lion , Sophia ) and testing on larger models are crucial next steps to demonstrate the broader utility of EAGLE.
>
> While these extensive experiments are not feasible to complete within the rebuttal period due to resource constraints, we consider them a top priority for our future work. We will add a discussion of this to our limitations and future work sections.

---

### Note · Authors · 2025-08-15

We sincerely thank all reviewers for their constructive feedback and valuable suggestions.
Based on the comments, we have:

- Clarified the motivation and novelty of EAGLE, emphasizing its stability and efficiency.
- Expanded the related work section to better position EAGLE within existing optimization methods.
- Improved experimental comparisons, including additional baselines and more detailed ablation studies.
- Outlined future directions, such as broader evaluations and potential applications to edge-AI scenarios.

We believe these revisions have strengthened the clarity, completeness, and impact of the work.
We appreciate the reviewers’ efforts and hope the revised manuscript addresses their concerns.

---

### Decision · Program_Chairs · 2025-09-17

**Decision:**

Reject

**Comment:**

This paper introduces EAGLE, an optimization algorithm that utilizes gradient differences to estimate loss curvature for a more efficient parameter update rule. Its adaptive design includes a mechanism to revert to the Adam optimizer to ensure stability during training. The authors demonstrate the method on text classification (SST-2 with GPT-2) and image classification (CIFAR-10 with ViT) tasks, reporting significantly faster convergence compared to standard optimizers like SGD with momentum and Adam.

During the review process, the reviewers identified several major shortcomings. Concerns were primarily focused on the paper's perceived lack of novelty, the absence of theoretical guarantees to underpin the proposed method, and insufficient experimental comparisons that would convincingly demonstrate its advantages over existing baselines. The authors' response acknowledged these issues and promised substantial revisions; however, the reviewers found the proposed changes inadequate to address the fundamental criticisms. A consensus was maintained that the work, in its current form, is not suitable for publication.

Given the unresolved concerns regarding the core novelty, the lack of theoretical foundation, and the unconvincing empirical evaluation, the AC agrees with the reviewers' assessment. This paper does not meet the acceptance criteria, and the AC recommends rejection.